# The Matricellular Protein Hevin Is Involved in Alcohol Use Disorder

**DOI:** 10.3390/biom13020234

**Published:** 2023-01-25

**Authors:** Amaia Nuñez-delMoral, Paula C. Bianchi, Iria Brocos-Mosquera, Augusto Anesio, Paola Palombo, Rosana Camarini, Fabio C. Cruz, Luis F. Callado, Vincent Vialou, Amaia M. Erdozain

**Affiliations:** 1Department of Pharmacology, University of the Basque Country, UPV/EHU, 48940 Leioa, Spain; 2Centro de Investigación Biomédica en Red de Salud Mental (CIBERSAM), Spain; 3Department of Pharmacology, Universidade Federal de São Paulo-UNIFESP, São Paulo 04023-062, Brazil; 4Department of Pharmacology, Instituto de Ciências Biomédicas, Universidade de São Paulo, São Paulo 05508-000, Brazil; 5Biocruces-Bizkaia Health Research Institute, 48903 Barakaldo, Spain; 6Institute of Biology Paris Seine, Neuroscience Paris Seine, CNRS UMR8246, INSERM U1130, Sorbonne Université, 75005 Paris, France

**Keywords:** hevin, astrocytes, alcohol use disorder, human post-mortem brain, Sparcl1

## Abstract

Astrocytic-secreted matricellular proteins have been shown to influence various aspects of synaptic function. More recently, they have been found altered in animal models of psychiatric disorders such as drug addiction. Hevin (also known as Sparc-like 1) is a matricellular protein highly expressed in the adult brain that has been implicated in resilience to stress, suggesting a role in motivated behaviors. To address the possible role of hevin in drug addiction, we quantified its expression in human postmortem brains and in animal models of alcohol abuse. Hevin mRNA and protein expression were analyzed in the postmortem human brain of subjects with an antemortem diagnosis of alcohol use disorder (AUD, *n* = 25) and controls (*n* = 25). All the studied brain regions (prefrontal cortex, hippocampus, caudate nucleus and cerebellum) in AUD subjects showed an increase in hevin levels either at mRNA or/and protein levels. To test if this alteration was the result of alcohol exposure or indicative of a susceptibility factor to alcohol consumption, mice were exposed to different regimens of intraperitoneal alcohol administration. Hevin protein expression was increased in the nucleus accumbens after withdrawal followed by a ethanol challenge. The role of hevin in AUD was determined using an RNA interference strategy to downregulate hevin expression in nucleus accumbens astrocytes, which led to increased ethanol consumption. Additionally, ethanol challenge after withdrawal increased hevin levels in blood plasma. Altogether, these results support a novel role for hevin in the neurobiology of AUD.

## 1. Introduction

Alcohol use disorder (AUD) is one of the psychiatric diseases that most negatively affect global health, society, and the economy [1,2]. Ethanol is considered the most dangerous and harmful drug of abuse, with an estimated 5.1% of the total disease burden (in terms of disability-adjusted life years) and 5.3% of deaths worldwide [3,4]. AUD is characterized by the compulsion to take or seek alcoholic beverages, craving, withdrawal symptoms, tolerance and relapse [5,6]. Chronic ethanol consumption alters the activity and the long-term function of brain regions such as the prefrontal cortex (PFC), hippocampus (HIP), striatum, amygdala (AMY) or cerebellum (CB), which result in cognitive impairments, learning and memory deficits, and a shift from positive to negative reinforcement [7,8]. An essential aspect of AUD is the high rate of relapse, depicted as recurrent abstinence periods followed by reinstatement of consumption. This pathological pattern of alcohol consumption might take part in the progression and deepening of the disorder [9,10,11]. Ethanol’s effects on the brain involve alterations in synaptic plasticity and morphological reorganization of circuits implicated in reward, stress and executive functions. Although the contribution of various molecular mediators, including monoamines, neuropeptides, and stress hormones, has been recognized [8], no key genetic switch mediating significant traits of the AUD phenotype has been characterized.

Astrocytes play critical roles in central nervous system homeostasis and are implicated in the pathogenesis of neurological and psychiatric conditions, including drug dependence [12,13]. In regard to AUD, activation of mouse cortical astrocytes by chemogenetic approaches increases alcohol consumption and alters its locomotor and sedative effects [14]. Likewise, activation of astrocytes in the nucleus accumbens (NAc) potentiates the rewarding properties of alcohol and decreases alcohol self-administration after prolonged abstinence [15]. Chronic alcohol exposure profoundly modifies the transcriptomic profile of astrocytes in vitro, in utero and in vivo, with changes in the expression of genes involved in cellular differentiation, metabolism, gene transcription, extracellular matrix composition and synaptogenesis [16,17,18,19]. Astrocytic synaptogenic factors promote excitatory synapse formation, maturation, and refinement during development [12,20,21,22,23,24]. In particular, hevin, a matricellular protein with synaptogenic properties secreted by astrocytes, is induced in prenatal and postnatal animal models of AUD [18,25].

Hevin (also known as SPARC-like 1 [SPARCL1], SC1, ECM2 or Mast9) specifically promotes the formation and maintenance of thalamocortical excitatory synapses, by linking presynaptic neurexin-1α and postsynaptic neuroligin-1B [21,22,24,26]. It is highly expressed in human and mouse adult brain [27,28,29,30,31,32], in astrocytes and parvalbumin interneurons and in a restricted number of other neuronal subpopulations, including glutamatergic neurons in cortical and subcortical regions [30]. Importantly, hevin has been implicated in resilience to stress, showing an antidepressant-like effect in a model of social defeat [33].

Given the effects of alcohol on the limbic system, and the role of hevin in structural and synaptic plasticity, we explored the role of hevin in AUD. The pattern of hevin expression was assessed in human cortico-limbic structures associated with AUD in postmortem brains of AUD subjects. We also assessed hevin levels in the brain and blood of mice exposed to different regimens of intraperitoneal alcohol administration. In addition, we evaluated the contribution of astrocytic hevin in the nucleus accumbens to ethanol preference and consumption by downregulating its expression using RNA interference.

## 2. Materials and Methods

### 2.1. Human Samples

Human samples were obtained at autopsy in the Basque Institute of Legal Medicine (Bilbao, Spain). Brain dissections were performed at the moment of the autopsy for the collection of PFC, HIP, caudate nucleus (CAU) and CB samples, which were immediately stored at −70 °C until assay. A blood toxicological study by the National Institute of Toxicology (Madrid), using different standard methods (including radioimmunoassay, enzymatic immunoassay, high-performance liquid chromatography and gas chromatography-mass spectrometry) determined the presence or not of ethanol, antidepressants, antipsychotics and psychotropic drugs at the time of death. The study included 75 subjects divided into three experimental groups: 25 subjects with an antemortem diagnosis of AUD, 25 subjects with an antemortem diagnosis of major depression and 25 control subjects, with no antemortem history of any neuropsychiatric disease. The antemortem diagnoses were performed by psychiatrists based on DMS-IV and ICD-10 criteria. The depression group was included as a control of disease-specific changes. Each subject with AUD was matched by age, sex and postmortem delay with a control and depression subject; none of these parameters were statistically different between groups (*p* > 0.05, one-way ANOVA). A full description of the demographic characteristics of all the subjects is shown in Appendix A. Samples from matched AUD, depression and control subjects were processed and tested in parallel. The study was performed in compliance with legal policy and ethical review boards for postmortem brain studies.

### 2.2. Animals and Treatments

All experiments were performed in strict conformity with the European Union laws and policies for use of animals in neuroscience research (European Committee Council Directive 2010/63/EU). C57BL/6J mice were housed in groups of five animals per cage under standard conditions of 23 °C and a 12 h light/dark cycle, with food and water provided *ad libitum*. Experiments were performed on 2 to 4-month-old mice.

Mice were exposed to four different ethanol regimens [34]. They were injected intraperitoneally (i.p.) either with saline or ethanol for 13 days, not injected for the following 3 days (withdrawal), and then given a challenge dose of saline or ethanol (relapse-like drinking). The four resulting experimental groups were as follows: (1) saline–saline (S–S), (2) saline–ethanol (S–E), (3) ethanol–saline (E–S) and (4) ethanol–ethanol (E–E). A rewarding dose of ethanol, 1.75 g ethanol/kg or 0.9% NaCl solution was used [35].

Mice were killed 24 h after the last injection, and brain dissections were performed immediately after. Five brain areas were collected with bilateral punches: frontal cortex (FC), amygdala (AMY), hippocampus (HIP), dorsal striatum (CPu) and nucleus accumbens (NAc). Blood was collected in EDTA-coated tubes and centrifuged to obtain plasma. All the samples were stored at −70 °C until their preparation for Western blot assays.

For hevin downregulation in astrocytes of the NAc, mice were anesthetized with an i.p. mixture containing 100 mg/kg of ketamine and 10 mg/kg of xylazine, placed in a stereotaxic frame, and injected with the adenovirus AAV2.5-GFAP-EmGFP-miR-hevin (9.4 × 10^12^ vg/mL) or the control viral vector AAV2.5.GFAP.eGFP.WPRE.hGH (9.1 × 10^12^ vg/mL, University of North Carolina, Chapel Hill, NC, USA). Coordinates (with respect to bregma) used to target the NAc were anteroposterior +1.6, mediolateral +1.45, and dorsoventral −4.3, with an angle of 10° from the midline. The virus was injected at a volume of 0.5 µL, at a rate of 0.1 µL/min. Behavioral tests were performed 3 weeks after surgery. The correct injection site and the expression of the transgene were confirmed by immunohistochemistry.

### 2.3. Sample Preparation

For Western blot experiments both human and animal brain samples were processed as previously described [31], with the following modifications: (1) in human brain samples the nuclear fraction was removed for all of the total homogenates, and (2) in mice brain punches the homogenization was performed by sonication (QSONICA Q55), due to the small sample volume. Protein content was measured by the Bradford method [36] and the samples were stored at −70 °C until analyzed by Western blot. Mouse plasma samples were homogenized in buffer supplemented with protease and phosphatase inhibitors (5 mM Tris-HCl pH 7.4, 50 µL/g of Sigma protease inhibitor cocktail, 5 mM Na_3_VO_4_ and 10 mM NaF) before sonication (QSONICA Q55) and stored at −70°C until analyzed by Western blot.

### 2.4. Real-Time Reverse Transcription Polymerase Chain Reaction (RT-qPCR)

mRNA extraction, cDNA synthesis and qPCR of the four human brain regions were performed as previously described [37] with minor modifications. The qPCR reactions were performed on a StepOne^TM^ system using TaqMan^TM^ Fast Universal Master Mix (Thermo Fisher) and the specific gene probes for hevin and the housekeeping genes glyceraldehyde 3-phosphate dehydrogenase (*GAPDH*) and ribosomal protein S13 (*RPS13*) (TaqMan^®^ gene expression assay Hs00949886_m1, Hs99999905_m1 and Hs01945436_u1, respectively). Hevin mRNA expression was normalized to the expression of the *GAPDH* and *RPS13* housekeeping genes and with a reference sample (pool of all samples for each brain region); and determined by the comparative Ct method (ΔΔCt), where ΔΔCt = (Ct (hevin)_sample_ − Ct (housekeeping)_sample_) − (Ct (hevin)_reference sample_ − Ct (housekeeping) _reference sample_) using StepOne^TM^ Software v2.1. A negative internal control (mQH_2_O) was also included. The qPCRs were run in triplicates for each cDNA.

### 2.5. Western Blot

Western blot experiments were performed as previously described [31]. In human and mouse brain samples hevin immunoreactivity was normalized against the β-actin signal. For mouse plasma samples, the Ponceau S staining confirmed the equal amount of protein load. In all cases, a standard pool of total homogenate (from human or mouse origin, respectively) was used as an external reference sample. The primary and secondary antibodies used for the detection of hevin and β-actin proteins are shown in Appendix A.

### 2.6. Intermittent Ethanol Access Schedule (IEA) in Hevin Knockdown Mice

In order to measure the alcohol intake in the active period of the mice, one week before starting the experiment, the light cycle was changed, with lights off at 9 a.m. and on at 9 p.m., and both hevin knockdown (KD, *n* = 7) and control (*n* = 7) mice were individually housed, with water and food *ad libitum*. Mice were first habituated to two bottles of water for 3 days. IEA was performed for a total of 18 days [38]; water and ethanol bottles were placed in each cage every day at 9 a.m. and removed at 5 p.m. The consumed volume was measured after 4 h and 8 h of the beginning of the experiment, and the bottles’ position was switched every day to avoid possible bottle side preference bias. Ethanol concentration (*w*/*v*) was increased every 3 days in order to induce an alcohol intake behavior. The initial ethanol concentration was 5%, followed by 10%, 15%, 20%, 3 days of ethanol withdrawal, and 10% of ethanol as the reinstatement concentration. Ethanol solutions were prepared by mixing 96% (*v*/*v*) ethanol with tap water. Mice were weighed every 3 days. Ethanol preference (percentage of ethanol consumed from the total volume for each measurement; >50% was considered as a preference for ethanol) and consumption (g ethanol/kg mouse ingested for each measurement) were calculated for each mouse.

### 2.7. Immunohistochemistry

Mice were anesthetized with a lethal dose of pentobarbital at 150 mg/kg i.p. and perfused intracardially with 4% paraformaldehyde in a phosphate-buffered solution. Brains were removed and stored at 4 °C in 4% paraformaldehyde until use. Brains were cut on a vibratome in coronal sections of 40 µm. Slices were incubated in a blocking buffer containing 0.2% gelatin and 0.25% triton in PBS for 1 h at room temperature, then treated with antibodies against hevin (mSPARCL1 AF2836, 1:2000, R&D) or anti-GFP (A010-pGFP-5, 1:2000, Badrilla) overnight at 4 °C. After washing, slices were incubated with a Cyanine Cy™3 conjugated secondary antibody for 2 h at room temperature (Jackson Immunoresearch, West Grove, PA, USA). Slices were incubated with DAPI (4’,6-diamidino-2-fenilindol, D9542, 1:15,000, Sigma-Aldrich, St. Louis, MO, USA) for 20 min at room temperature before mounting with Fluoromount-G. All the slices were scanned on a NanoZoomer 2.0-HT (Hamamatsu Photonics, Hamamatsu City, Japan) for quantification, as previously described [30].

### 2.8. Statistical Analysis

Statistical analysis was performed using GraphPad Prism 9^®^. Comparison of hevin mRNA and protein levels in the human brain between the AUD, depression and control groups was performed using one-way ANOVA followed by Dunnett’s multiple comparisons test. Taking into account the large sample size (*n* = 25) and the high inter-subject variability in human brain morphology and configuration, data were considered normally distributed [39,40]. Data obtained in animal studies did not pass the D’Agostino and Pearson normality test and F-test for the equality of variances, so non-parametric statistical tests were used. Specifically, biochemical analyses in mouse brains and plasma were performed using the Kruskal-Wallis test followed by Dunn’s multiple comparisons test. Differences between hevin KD mice and controls in ethanol preference and consumption were assessed with the Mann-Whitney test and the Friedman test was used for ethanol concentration data. Results were considered statistically significant when the *p*-value was <0.05.

## 3. Results

### 3.1. Hevin Expression in Postmortem Human Brains of Subjects with AUD

Hevin mRNA and protein expression levels were measured in four human brain areas (PFC, HIP, CAU and CB) from 75 subjects. Individuals were divided according to their antemortem psychiatric history: AUD (*n* = 25), depression (*n* = 25) and control (*n* = 25) groups. Interestingly, in all the studied brain areas an increase in hevin mRNA was detected in the AUD group compared to the controls (PFC: 59% increase, *p* = 0.0028; HIP: 72% increase, *p* = 0.0058; CAU: 92% increase, *p* = 0.0028 and CB: 70% increase, *p* = 0.0344). No significant differences were detected between the depression and control groups (Figure 1A–D).

To further elucidate whether hevin mRNA increase paralleled an enhancement in the translation process, the protein expression levels were also determined. Both previously described ~130 kDa and ~100 kDa hevin bands [22,31,41,42] were measured in total homogenates of PFC, HIP, CAU and CB. In accordance with the alterations observed in hevin mRNA, an increase in the ~130 kDa and ~100 kDa forms was detected in PFC and CB of AUD subjects in comparison to controls (PFC: 25% increase in ~130 kDa band, *p* = 0.0015 and 46% increase in ~100 kDa band, *p* < 0.0001; CB: 32% increase in ~130 kDa band, *p* = 0.0344 and 38% increase in ~100 kDa band, *p* = 0.013) (Figure 2A,D). In the HIP, only the ~100 kDa band was significantly increased in AUD (56% increase, *p* = 0.00384), although the ~130 kDa band showed a similar trend (Figure 2B). In contrast, in the CAU, no significant differences were observed between both groups in the expression of any of the bands (Figure 2C). No correlation was found between ethanol levels detected in the blood at the time of death and levels of brain hevin mRNA or protein expression. Finally, the depression group did not show significant differences in the levels of protein expression in comparison to the control group in any brain area, as observed for the mRNA levels (Figure 2A–D).

### 3.2. Hevin Protein Expression in Mouse Brain after Ethanol Administration

Modeling the complex facets of AUD in animals is challenging. Repeated i.p. administration of ethanol in mice induces behavioral sensitization, a model associated with relapse to drug-seeking behavior [43]. Here, hevin expression was evaluated in several mice brain regions (FC, AMY, HIP, CPu and NAc) after different ethanol regimens. To control ethanol plasma levels, mice were injected i.p. with ethanol (13 days) followed by a withdrawal period (3 days) and a challenge dose of ethanol or saline. In contrast to the human brain, mice did not express the ~100 kDa band, or its immunoreactive signal was too low to be analyzed, as described before [42,44]. Therefore, the present data correspond only to the ~130 kDa band. Twenty-four h after an ethanol challenge in animals injected repeatedly with ethanol, an increase in hevin protein expression was observed in the NAc in comparison to the other three treatments (saline, one single dose of ethanol and withdrawal period after repeated injections) (74% increase vs S–S, *p* = 0.0041; 68% increase vs S–E, *p* = 0.0232, and 98% increase vs E–S, *p* = 0.0007) (Figure 3A). Acute ethanol injection reduced hevin protein levels in the AMY (54% decrease vs S–S, *p* = 0.0471 (Figure 3B). No significant changes were observed in the other brain regions.

### 3.3. Hevin Protein Expression in Mouse Plasma after Ethanol Treatment

Hevin expression levels were also analyzed in mouse plasma after chronic ethanol treatment. The ~100 kDa band was not identifiable in these samples. Ethanol challenge after withdrawal produced higher levels of the hevin ~130 kDa band in comparison to the saline challenge after withdrawal (65% increase vs E–S, *p* = 0.0396) (Figure 4). An additional band with a molecular weight of around 47 kDa was also analyzed due to the high immunoreactivity signal of this band in most of the samples. Plasma showed high variability in the expression levels of the 47 kDa band among mice, with some of them expressing high levels of this form (i.e., two mice from the E–S group), and almost undetectable levels for others. As a result, no differences were detected in hevin expression levels between the ethanol treatments for the 47 kDa band (Figure 4). In order to assess if the expression of the lower band was the result of the cleavage of the 130 kDa band, the ratio of the immunoreactivity of both bands was also calculated (130 kDa/47 kDa). No significant differences were detected between treatment groups in the 130 kDa/47 kDa ratio (Figure 4).

### 3.4. Behavioral Consequences of Hevin Downregulation in NAc Astrocytes on Ethanol Consumption

To test the role of hevin in the long-term adaptation to ethanol, we downregulated hevin expression in mouse NAc astrocytes using an RNA interference strategy and tested the consequences of this cell-specific and regional downregulation of hevin on the consumption of ethanol in an intermittent ethanol access schedule (IEA). The downregulation of hevin in the astrocytes of the NAc was verified by immunohistochemistry of hevin protein and GFP in brain slices after the IEA schedule was completed. Immunohistochemistry assays confirmed a complete deletion of hevin expression in NAc astrocytes from mice infected with the miRNA-containing viruses (Figure 5A).

Mice showed a high preference for the initial low dose of the ethanol solution (Figure 5B). Preference for the ethanol bottle (calculated as the percentage of ethanol consumed from the total volume) was similar between hevin KD mice and control mice for every dose during the entire procedure (Figure 5B). When measuring total alcohol consumption (ingested ethanol expressed as g ethanol/kg), hevin downregulation in NAc astrocytes did not affect consumption of ethanol at low and medium concentrations (5%, 10% and 15%, *w*/*v*) (Figure 5C). However, when exposed to a higher concentration of ethanol (20%), hevin KD mice showed an increased consumption compared to the lowest dose (5%), while control animals maintained a similar intake over time (Figure 5C). After three days of withdrawal, hevin KD mice showed a higher relapse to consuming 10% ethanol than controls (Figure 5C). Interestingly, these effects were only observed in the first 4 h of ethanol access. The last 4 h of the IEA experiment (from 4 h to 8 h) revealed that both groups, hevin KD and control mice, drank as much water as ethanol. Altogether, these data demonstrate that hevin KD animals consume more ethanol at the highest concentration and after ethanol withdrawal while having a similar preference for ethanol compared to control mice.

## 4. Discussion

AUD is characterized by two main features, the loss of control over the consumption of alcoholic beverages, and the relentless necessity to consume alcohol during abstinence, known as craving [5,6]. However, their neurobiological bases remain mainly unknown. Postmortem brain samples from subjects with a reliable antemortem diagnosis of AUD, and appropriately matched control subjects, provide a unique opportunity to search for any clinical relevance of the altered molecular expression of different proteins in patients with AUD [45]. In this line, the principal finding of the present study is that the matricellular protein hevin is overexpressed in the brain of subjects with AUD.

Human brain regions analyzed in this study were selected based on their implication in different aspects of AUD and on tissue availability. The PFC was chosen for its role in executive functions, such as control, planning and inhibition of behaviors; the HIP for its important role in the association of particular environments and emotional states with the effect of the drug; the CAU, for its involvement in cognition, learning, memory, and feedback processing; and the CB, for its role in movement and balance incoordination after alcohol consumption, in addition to its role in cognition, emotion and drug addiction [8]. All the studied regions showed an increase in hevin levels either at the mRNA and/or protein levels in AUD subjects. This is the first evidence in humans for the role of hevin in the pathophysiology of AUD. Even if the mechanisms underlying this observation or its consequences are difficult to determine in humans, several hypotheses can be postulated. Hevin promotes the formation and maintenance of excitatory synapses specifically and modulates the refinement and stabilization of dendritic spines [22,24,26,46]; therefore, the upregulation of hevin expression could be implicated in the profound structural synaptic rearrangement/adaptation that occurs in AUD [47,48]. Indeed, we previously found an important loss in neuronal structural proteins in AUD subjects, and other authors have reported that AUD subjects also suffer neuronal and dendritic spine loss [48,49,50,51,52]. The increase in hevin could thus be an adaptative mechanism triggered to counteract the effects of ethanol on neuronal structure. Furthermore, at a molecular level, hevin increases NMDA receptor expression and function [21,24,46]. Hence, hevin could also mediate part of the synaptic plasticity occurring after chronic ethanol, such as long-term potentiation and NMDA receptor-dependent plasticity, described in preclinical studies as well as in postmortem human brains from subjects with AUD [47,53,54,55]. In our study, we did not find any difference in hevin expression levels between sexes. As a control for disease specificity, a group of subjects with antemortem diagnoses of depression was added. Hevin expression was unaltered in the depression group in our study, consistent with a previous study [56]. These results in the human brain raise the question of whether hevin overexpression is induced by ethanol or if it is a pre-AUD state, suggesting vulnerability.

We used preclinical models of AUD and compared the effects of chronic alcohol withdrawal, chronic alcohol relapse and acute alcohol administration on hevin protein levels in mouse brain. We selected cortico-limbic brain regions involved in the neurocircuitry of drug addiction, in particular ventral and dorsal striatum, PFC, HIP and AMY [8]. Because the cerebellum has been classically associated with movement and balance incoordination after long-term alcohol consumption, we did not include this region in our mouse model. In contrast to human data, we observed hevin protein expression changes only in the AMY and NAc. In the AMY, a decrease in hevin levels was observed after a single dose of ethanol, but not after chronic ethanol administration. In this context, and according to the possible role of hevin in regulating glutamatergic signaling and neuroplasticity in response to ethanol, acute and low doses of ethanol decrease excitability and increase dendritic spines in the rodent AMY, coinciding with anxiolytic effects, while repeated withdrawal normalizes dendritic arborization, causing tolerance to ethanol anxiolysis [57,58,59]. In contrast, in the NAc, hevin was overexpressed only with a challenging dose after withdrawal, while no change was observed after acute ethanol injection. Chronic ethanol intake and withdrawal produce alterations in the rodent NAc in dendritic spine number, structure and organization, and increase NMDA and AMPA receptor expression and function, which suggests that the increase in hevin observed in the NAc could be a compensatory response to the ethanol-related glutamatergic inhibition [60,61,62,63,64,65]. Overall, while changes in hevin levels in the AMY could be implicated in the acute alcohol experience, the increased hevin expression in the NAc after chronic ethanol exposure suggests that hevin might also participate in the long-term adaptation to ethanol. As noted, mouse and human brains showed a different pattern of hevin expression in several regions (i.e., PFC, HIP) after alcohol exposure. This difference could be explained by two essential factors largely discussed in the literature: (1) the different regimens of alcohol exposure (time course, total drug intake and probable withdrawal period) which raises the question of what the necessary cumulative effect of ethanol is to cause neuroplasticity, and (2) different psychosocial and environmental stressors [66].

To test the potential role of hevin on alcohol consumption, we used an RNA interference strategy to downregulate hevin expression in NAc astrocytes, the main cell type expressing hevin [30], and evaluated the consequences of hevin manipulation in the IEA model of voluntary ethanol intake. Our results indicate a disrupted reward system when hevin is downregulated. Specifically, we detected a higher ethanol consumption in hevin KD mice in two cases: (i) at high ethanol concentrations (20%), which were administered after 15 days of intake, and (ii) after three days of withdrawal with an intermediate ethanol concentration (10%). These findings suggest that ethanol addiction and withdrawal are associated with dysregulated hevin signaling. Since hevin increases excitatory synaptic plasticity, knocking-down hevin might disrupt the glutamatergic plasticity involved in the rewarding effects of ethanol [67], making it necessary to increase alcohol consumption in order to reach the same hedonic response [68]. Nevertheless, the ethanol preference analysis revealed that both hevin KD and control mice drank as much water as ethanol, showing no differences in the preference at any dose or time. We suggest that mice undergo autoregulation in the water intake probably to avoid the dehydration produced by the higher ethanol intake. Despite the inherent limitations of experimental models of psychiatric disorders, the present model contributes to a better understanding of the underlying plasticity of alcohol relapse.

Finally, we tested whether hevin levels were altered in the plasma of mice after chronic alcohol withdrawal, chronic alcohol relapse or acute alcohol administration. Interestingly, a very similar expression profile was obtained after the different ethanol treatments in plasma samples compared to the NAc, with increased hevin expression only with a challenge dose after withdrawal. The origin of hevin protein in plasma is still uncertain. Hevin is not expressed by any blood cell type [31], but is expressed in other tissues than the brain, such as the lung [69], endothelial cells of high endothelial venules [27,70] and the stomach [71]. The increase in hevin in the plasma of ethanol-treated mice suggests that hevin may be a component of the response involved in remodeling events associated with neuronal degeneration following neural injury, as has been proposed in neurodegenerative diseases and several cancers [72,73]. In animal models of epilepsy, ischemia and injury, hevin is induced in reactive astrocytes, suggesting that it might participate in neuronal remodeling in pathological conditions in the adult stage [74,75].

As a whole, this study sheds light on the role of hevin in the pathophysiology of AUD. In humans, we previously found that hevin is expressed in astrocytes and some glutamatergic and GABAergic neurons [30]. The source of hevin overexpression in AUD subjects is unknown. Our cell-specific RNA interference strategy in mice shows that hevin expression in NAc astrocytes is crucial for increased voluntary ethanol consumption. As described above, we hypothesize that hevin could take part in the acquisition of the addiction behavior by strengthening and promoting specific synaptic signaling and/or modulating neuronal plasticity/adaptation in brain areas involved in AUD.

In conclusion, our postmortem brain analysis revealed an increase in hevin levels in various brain regions in subjects with AUD. In order to explore whether this increase was a consequence of alcohol intake, we used a mouse model of chronic ethanol administration and found that only chronic ethanol relapse increases hevin in the brain reward center, the NAc. In addition, we evaluated whether alterations in hevin expression affected alcohol-related behavior in mice, and revealed that hevin downregulation in NAc astrocytes, where the vast majority of hevin is expressed in this region, modulates alcohol consumption. Overall, this study provides the first evidence for the role of hevin in alcohol consumption and addiction.

## Figures and Tables

**Figure 1 biomolecules-13-00234-f001:**
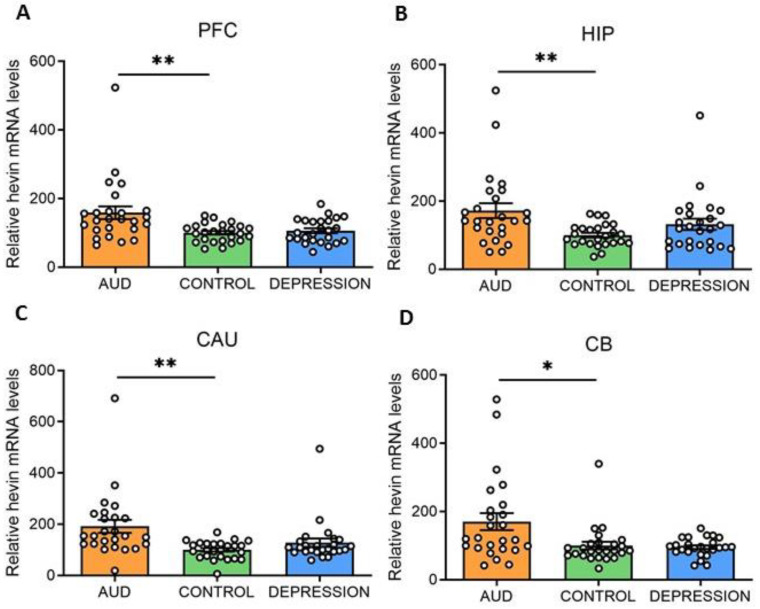
**Hevin mRNA expression in postmortem human brains of alcohol use disorder (AUD) subjects.** Relative hevin mRNA levels were quantified by RT-qPCR in postmortem brain samples from subjects with AUD, depression and controls in the prefrontal cortex (PFC) (**A**), hippocampus (HIP) (**B**), caudate nucleus (CAU) (**C**) and cerebellum (CB) (**D**). *n* = 25 per group. One-way ANOVA followed by Dunnett’s post-hoc test, * *p* < 0.05, ** *p* < 0.01. Samples were run in triplicate. Bars show mean ± SEM, dots represent the average value of each individual.

**Figure 2 biomolecules-13-00234-f002:**
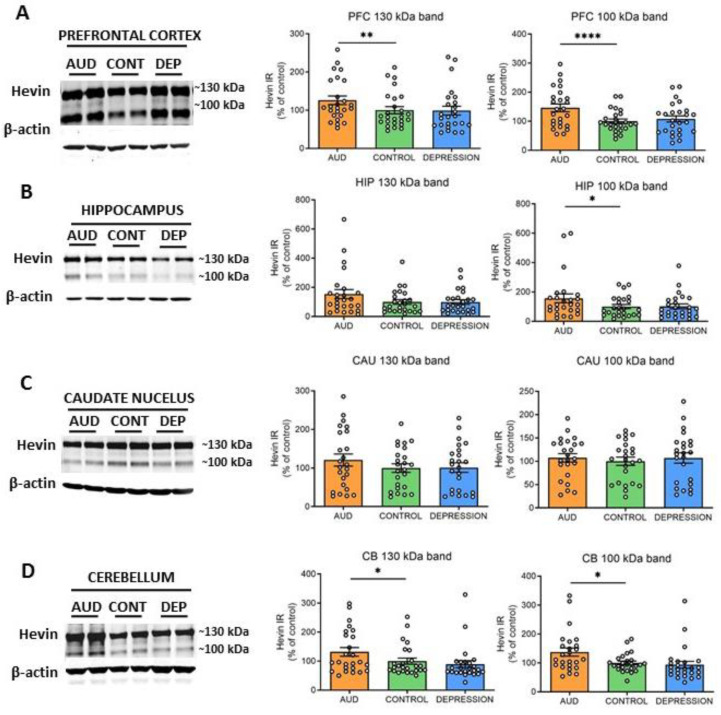
**Protein expression of hevin in postmortem human brains of alcohol use disorder (AUD) subjects.** Hevin protein expression in postmortem brain samples from subjects with AUD, depression (DEP) and controls (CONT) in the prefrontal cortex (PFC) (**A**), hippocampus (HIP) (**B**), caudate nucleus (CAU) (**C**) and cerebellum (CB) (**D**). Representative Western blot images for each brain region are shown, along with the graphics representing the actin-normalized immunoreactivity (IR) of both the ~130 kDa and ~100 kDa hevin bands. *n* = 25 per group. One-way ANOVA followed by Dunnett’s post-hoc test, * *p* < 0.05, ** *p* < 0.01, **** *p* < 0.0001. Experiments were performed in triplicate. Bars show mean ± SEM, dots represent the average value of each individual.

**Figure 3 biomolecules-13-00234-f003:**
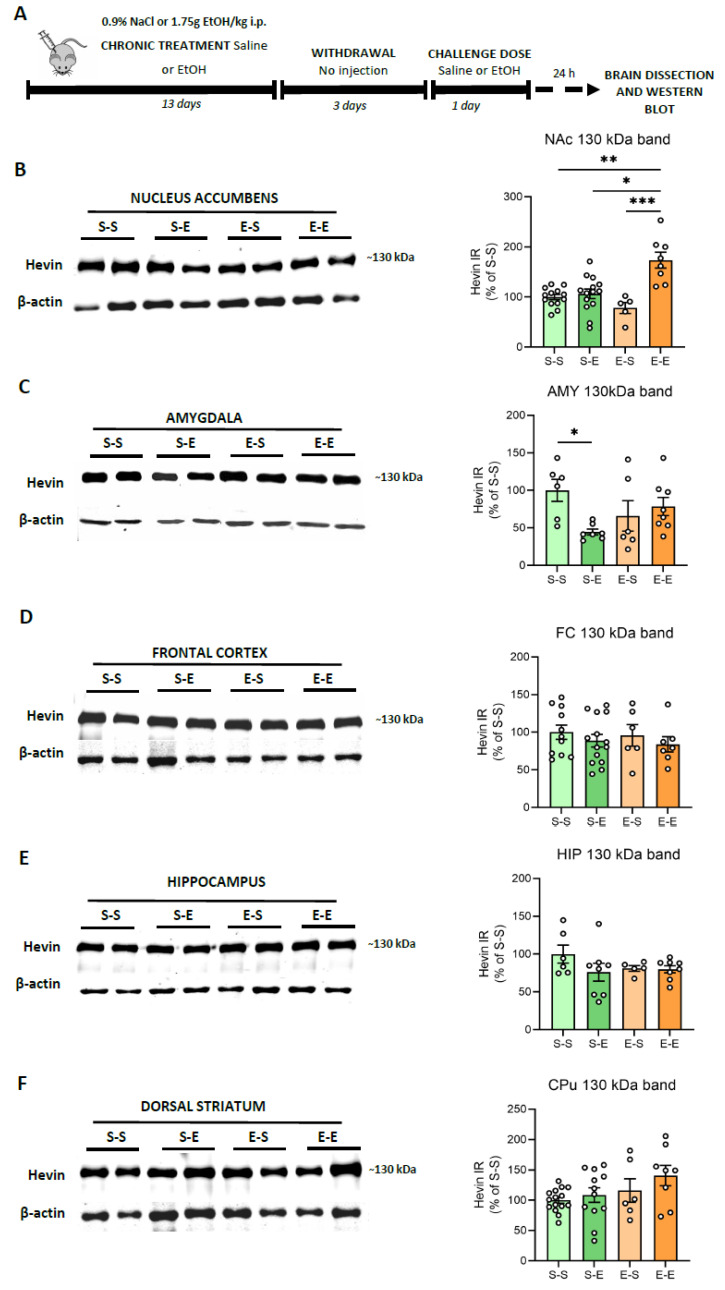
**Hevin protein expression in the brain of mice exposed to different regimens of ethanol.** (**A**) Schematic figure showing the four ethanol treatments performed. (**B**–**F**) Representative Western blot images and the graphics representing the actin-normalized immunoreactivity (IR) of the ~130 kDa band in the four treatment groups (S–S: saline–saline, S–E: saline–ethanol, E–S: ethanol–saline and E–E: ethanol–ethanol) in the following brain regions: (**B**) nucleus accumbens (NAc), (**C**) amygdala (AMY), (**D**) frontal cortex (FC), (**E**) hippocampus (HIP) and (**F**) dorsal striatum (CPu). *n* = 5–15 per group. Kruskal-Wallis followed by Dunn’s post-hoc test, * *p* < 0.05, ** *p* < 0.01, *** *p* < 0.001. Experiments were performed in triplicate. Bars show mean ± SEM, dots represent the average value of each individual.

**Figure 4 biomolecules-13-00234-f004:**
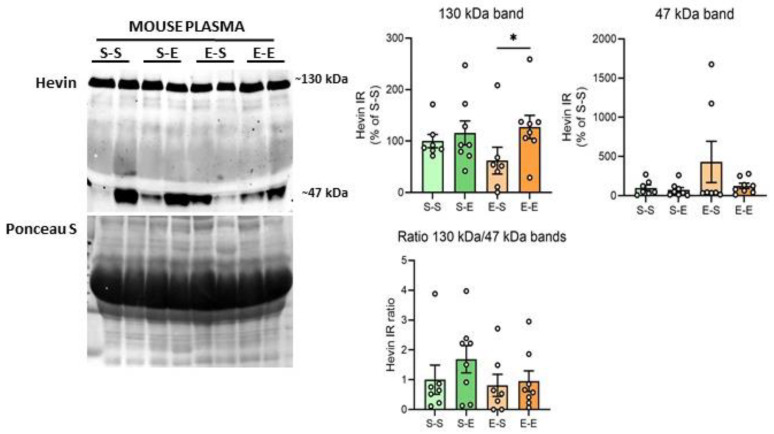
**Hevin protein expression in mouse plasma after different regimens of ethanol**. A representative Western blot of hevin in mouse plasma and its Ponceau S stain images, along with the graphics representing the actin-normalized immunoreactivity of the ~130 kDa and ~47 kDa hevin bands and the 130 kDa/47 kDa band ratio in the four treatment groups (S–S: saline–saline, S–E: saline–ethanol, E–S: ethanol–saline and E–E: ethanol–ethanol). *n* = 7–8 per group. Kruskal-Wallis followed by Dunn’s post-hoc test, * *p* < 0.05. Experiments were performed in triplicate. Bars show mean ± SEM, dots represent the average value of each individual.

**Figure 5 biomolecules-13-00234-f005:**
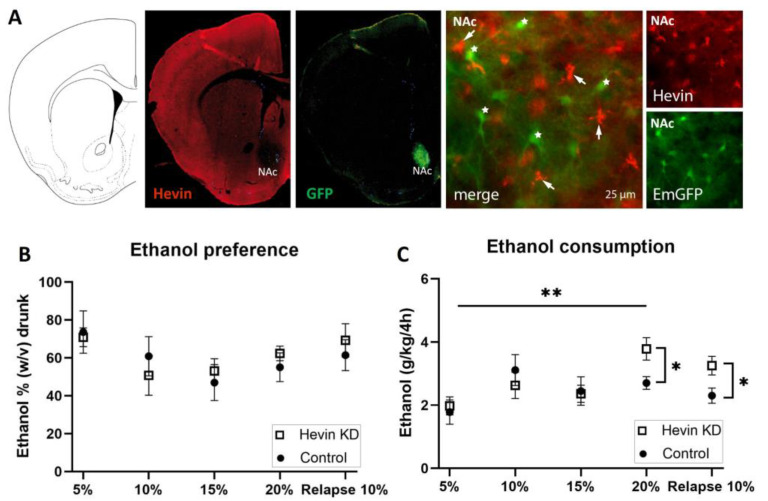
**Effect of hevin protein downregulation in nucleus accumbens (NAc) astrocytes on ethanol consumption in mice.** (**A**) Immunohistochemical validation of hevin downregulation in NAc astrocytes after injection of miRNA-containing viruses. Hevin immunofluorescence, in red, is absent in astrocytes infected with hevin miRNA and co-expressing EmGFP (star). Non-transfected astrocytes express hevin (arrow). (**B**) Ethanol preference quantification every third day of each ethanol concentration. (**C**) Ethanol consumption quantification every third day of each ethanol concentration. The ordinate represents the average of the percentage of ethanol taken by hevin KD and control groups (**B**) or the grams of ethanol per kg of mouse weight taken in 4 h (**C**). The abscissa shows the ethanol concentration tested. *n* = 8/group. Mann-Whitney test was used for the comparison of the preference and consumption between the hevin KD mice and controls (* *p* < 0.05, ** *p* < 0.01); and Friedman followed by Dunn’s post-hoc test was used for the comparison of the preference and consumption between ethanol concentrations. Values represent mean ± SEM.

## Data Availability

Not applicable.

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
