# Peer review of "The Matricellular Protein Hevin Is Involved in Alcohol Use Disorder"

_biomolecules, 2023, doi:10.3390/biom13020234_

Round 1
Reviewer 1 Report
In this article, Nuñez-del Moral & colleagues investigated an involvement of the matricellular protein, hevin, in alcohol use disorder (AUD) studying human postmortem brains, and in animal models of alcohol abuse. They showed that hevin plays a role in the neurobiology of AUD. The rationale and study design seem to be quite reasonable and the findings reported are interesting. However, before being accepted, I would suggest the authors clarify or provide more details regarding the following points:
1. Please provide more details on the mouse model of alcohol dependence used in the study. Please indicate the advantages and limitations of this model in relation to other models of alcohol dependence.
2. In the Discussion section, Authors should provide information on hevin's involvement in various brain injury processes, such as Alzheimer's type neurodegeneration, hypoxia, stroke, allodynia etc. Evidence exist that the neuron-glia interaction, with hevin engaged in, can be exploited to modulate synaptic reorganization under various neurological conditions. So, this may be a component of the 'response' involved in remodeling events associated with neuronal degeneration following neural injury. Thus, hevin should not be considered as a specific “biomarker of alcohol abuse” but rather an unspecific element of response to the neuronal degeneration that may be induced by various agents and processes including alcohol abuse analyzed in the current study. The discussion of the results should be modified in this respect.
Reviewer 2 Report
This study addressed the possible role of hevin in drug addiction and quantified Hevin mRNA and protein mRNA and protein expression in human postmortem brains and in animal models of alcohol abuse. They also test if this alteration was the result of alcohol exposure or indicative of a susceptible factor to alcohol consumption using an RNA interference strategy, which results suggesting that hevin could be used as a biomarker of alcohol abuse. The study provides a novel role for hevin in the neurobiology of AUD. The approaches and data appear reasonable and clear.
However, only one question: the siRNA knockdown experiment may not be totally able to that hevin could be used as a biomarker of alcohol abuse. Do you notice any known specific drugs to alter this hevin for AUD treatment? Please add one or two paragraphs to make a comparison for AUD treatment by siRNA and/or drugs in the discussion section.
Reviewer 3 Report
The paper is very interesting, with a focus on the role of the extracellular matrix protein hevin in the maintenance of astrocytes and synaptic plasticity in AUD. The Authors analyzed four cortico-limbic regions in the postmortem human brain in three experimental groups. On the one hand, the complexity of the human brain combined with the uniqueness of neuropsychiatric conditions means that research utilizing postmortem human brain tissue in a number of specific aspects has significant advantages compared to animal-based studies. On the other hand, the Authors designed their research to strengthen results found in humans using animal models. In mouse experiments, the Authors investigated hevin expression in several brain regions of mice after ethanol administration, as well as in the plasma of mice after ethanol treatment.
Nowadays, unfortunately, for various reasons, such complex studies are rare.
However, some questions are raised:
1-Why are the regions used in mice different from those used in humans? It would be interesting to compare the same regions, including the cerebellum.
2-Was ethanol discovered in the blood in Case 4 (controls)? I understand that this could have been just the one fatal ethanol use, but anyway, was AUD also confirmed at autopsy by evaluating changes in organs such as the heart, pancreas, and liver? Are there histological data on these organs and changes in them?
